# Antioxidant, Xanthine Oxidase, α-Amylase and α-Glucosidase Inhibitory Activities of Bioactive Compounds from *Rumex crispus* L. Root

**DOI:** 10.3390/molecules24213899

**Published:** 2019-10-29

**Authors:** Truong Ngoc Minh, Truong Mai Van, Yusuf Andriana, Le The Vinh, Dang Viet Hau, Dang Hong Duyen, Chona de Guzman-Gelani

**Affiliations:** 1Center for Research and Technology Transfer (CRETECH), Vietnam Academy of Sciences and Technology, Hanoi 10072, Vietnam; thevinh3839@gmail.com (L.T.V.); hauhoahock20@gmail.com (D.V.H.); hongduyen1908@gmail.com (D.H.D.); 2Graduate School for International Development and Cooperation (IDEC), Hiroshima University, Higashi-Hiroshima 739-8529, Japan; truongmaivan1991@gmail.com (T.M.V.); yusufandriana@yahoo.com (Y.A.); 3Research Center for Appropriate Technology, Indonesian Institute of Sciences, Subang, 41213, Indonesia; 4Department of Chemistry, College of Science and Mathematics, Mindanao State University - Iligan Institute of Technology, Iligan 9200, Philippines; chona.gelani@g.msuiit.edu.ph

**Keywords:** bioactive compound, anti-radical, anti-gout, anti-diabetic, *Rumex crispus*, chrysophanol, physcion

## Abstract

The root of *Rumex crispus* L. has been shown to possess anti-gout and anti-diabetic properties, but the compounds responsible for these pharmaceutical effects have not yet been reported. In this study, we aimed to isolate and purify active components from the root of *R. crispus*, and to evaluate their anti-radical, anti-gout and anti-diabetic capacities. From the ethyl acetate (EtOAc) extract, two compounds, chrysophanol (**1**) and physcion (**2**), were isolated by column chromatography with an elution of hexane and EtOAc at a 9:1 ratio. Their structures were identified by spectrometric techniques including gas chromatography-mass spectrometry (GC-MS), electrospray ionization-mass spectrometry (ESI-MS), X-ray diffraction analyses and nuclear magnetic resonance (NMR). The results of bioassays indicated that (**1**) showed stronger activities than (**2**). For antioxidant activity, (**1**) and (**2**) exhibited remarkable DPPH radical scavenging capacity (IC_50_ = 9.8 and 12.1 µg/mL), which was about two times stronger than BHT (IC_50_ = 19.4 µg/mL). The anti-gout property of (**1**) and (**2**) were comparable to the positive control allopurinol, these compounds exerted strong inhibition against the activity of xanthine oxidase (IC_50_ = 36.4 and 45.0 µg/mL, respectively). In the anti-diabetic assay, (**1**) and (**2**) displayed considerable inhibitory ability on α-glucosidase, their IC_50_ values (IC_50_ = 20.1 and 18.9 µg/mL, respectively) were higher than that of standard acarbose (IC_50_ = 143.4 µg/mL). Findings of this study highlight that (**1**) and (**2**) may be promising agents to treat gout and diabetes, which may greatly contribute to the medicinal properties of *Rumex crispus* root.

## 1. Introduction

*Rumex crispus* L. is known as the curly dock or yellow dock in most of Europe, North Africa, Turkey, Northern Iran, Central and East Asia, and North America [1]. The roots of this plant have been used in traditional medicine as a tonic, laxative, and for hemostasis medication. The fruits (seeds) are used for the treatment of dysentery. The young leaves of the plant are eaten as servings of mixed greens and soups. The plant has been utilized for treating the relevant disease, dermatology contaminations, gastrointestinal tract maladies, upper respiratory tract illnesses, and loose bowels [2,3]. The bioactive substances detected in this plant include flavonoids (isorientin, vitexin, orientin and isovitexin), lipids, vitamins, carotenoids, natural acids and minerals. The root of *R. crispus* may be a wealthy source of anthraquinones glycosides (chrysophanol and emodin) [4].

Antioxidant prevention agents are common or manufactured substances having the capacity to hinder or delay oxidation at relatively low concentrations [5]. Antioxidant compounds are known to combat oxidative stress. When oxidative stress is uncontrolled it is associated with several pathophysiological processes [6]. In a pharmacological examination of the ether, ethanol, and hot water extracts of *R. crispus*, the water extract showed the most elevated antioxidant activity. The most noteworthy amount of total phenolic compounds was found within the ethanol extract of the seeds. In regards to the reducing power and DPPH scavenging activity, the ethanol extract of the seeds was the foremost compelling [7]. The power to quench singlet oxygen and the protective effects of various extracts (hexane, chloroform, ethyl acetate and butanol) of *R. crispus* seeds against photodynamic damage were investigated in a biological system. A higher total phenolic content was observed for the ethyl acetate (EtOAc) and butanol extracts. The level of *in vitro* antioxidant action of the methanol concentrate of *R. crispus* was determined by measuring its ferric-lessening antioxidant activity, without DPPH radical searching movement and the ability to impact the lipid peroxidation in liposomes. It was seen that the methanol concentrate had direct antioxidant prevention agent movement. In view of the *in vivo* tests, it was inferred that the dose routine did not impact the degrees of lipid peroxidation. The methanol concentrate of *R. crispus* roots displayed DPPH radical searching (IC_50_ = 42.86 μg/mL) [7,8]. The ethereal piece of *R. crispus* can be utilized as a compelling and safe wellspring of antioxidant prevention agents. However, information on the anti-gout and anti-diabetic properties of the underground plant parts is still limited.

Xanthine oxidase (XOD) is an important enzyme responsible for hyperuricemia, and a predisposing factor for gout and oxidative stress-related diseases. This enzyme plays an important role in catalyzing the oxidation of hypoxanthine to xanthine and xanthine to uric acid [9,10]. Nowadays, few drugs are commonly used to treat gout including allopurinol and febuxostat. However, some serious and undesirable effects on skin caused by allopurinol and febuxostat may occur when taking the drug, including Steven-Johnson syndrome (SJS) and toxic epidermal necrolysis (TEN), causing death in up to 39% of cases. Lesions can occur on other organs such as the liver and kidney [11,12]. Many studies have reported that compounds possessing both antioxidant properties will be effective in gout treatment [13]. On the other hand, antioxidants are able to prevent or slow down oxidation by eliminating free radicals, which further help prevent oxidative disease and increase people’s lifespan [14]. On the other hand, xanthine oxidase enzyme inhibitors can reduce the enzyme activity, preventing the formation of urate salts [15,16].

Diabetes has become a worldwide health problem in developed and developing countries. It is reported that about 425 million people are suffering from diabetes, which accounts for 12% of global health expenditure [17]. Among diabetic types, the pathogenesis of type 2 diabetes may launch from many factors such as genetic predisposition, environment, and pancreatic beta-cell dysfunction [18]. Therefore, one of the diabetes treatments is to inhibit the enzyme activity of α-amylase and α-glucosidase to minimize the formation of blood glucose [19]. On the other hand, oxidative tension in diabetes coexists with a reduction in antioxidant status [20], which can increase the deleterious effects of free radicals. It has also been known that alloxan induces diabetogenic activity mainly by inducing oxygen free base and thereby damaging the pancreas [21]. Supplementation with non-toxic antioxidants may have a chemoprotective role in diabetes [22]. Antioxidants as well as vitamins C and E, have been shown to reduce oxidative stress in experimental diabetes [23]. Supplementation of vitamin C has also been shown to lower berth glycosylated hemoglobin in diabetic patients [23]. Many industrial plant excerpts and plant products have been shown to have significant antioxidant activity [24,25,26].

This study aims to isolate compounds from *R. crispus* root that can scavenge free radicals and inhibit xanthine oxidase, α-amylase and α-glucosidase. The results of the study will contribute to a better understanding of the value of medicinal plants, as well as contribute to the database for the production of gout and diabetes medications. Biological properties of the isolated compounds were also examined for their efficacy and safety in antioxidant activity and inhibition of xanthine oxidase, α-amylase and α-glucosidase enzyme by *in vitro* methods. The obtained compounds were identified by gas chromatography-mass spectrometry (GC-MS), electrospray ionization (ESI), atmospheric pressure chemical ionization (APCI), nuclear magnetic resonance (NMR), and X-ray diffraction.

## 2. Results

### 2.1. Structure Elucidation of Isolated Fractions

Separation of bioactive compounds from *Rumex crispus* root (RCR) was conducted following the procedure illustrated in Figure 1.

The isolated metabolites were characterized and identified by GC-MS, ESI-MS, APCI-MS, ^1^H- and ^13^C-NMR and X-ray analyses (Appendix A and Appendix A).

On the basis of GC-MS data (Table 1), isolated fractions were identified. The structure and formula of compounds were further confirmed by ESI-MS. Two fractions, **C2** and **C3**, were elucidated by ^1^H NMR and ^13^C NMR. Chemical structures of the identified compounds are illustrated in Figure 2 and Figure 3.

### 2.2. NMR Structural Elucidation

Compound **1** was isolated as an orange needle from the EtOAc extract of RCR. Its molecular formula, C_15_H_10_O_4_, was deduced from a combined analysis of the positive ESI MS at *m/z* 255.3 [M + H]^+^ (C_15_H_11_O_4_) and ^1^H-, ^13^C-NMR, HSQC, and HMBC spectra. The ^1^H-NMR spectrum of **1** showed the presence of a methyl group at δ_H_ 2.46 (3H, s, H-11) and five protons of aromatic ring at δ_H_ 7.82 (1H, dd, *J* = 7.0, 1.0 Hz, H-5), 7.67 (1H, t, *J* = 8.0 Hz, H-6), 7.64 (1H, s, H-4), 7.30 (1H, dd, *J* = 8.5, 1.0 Hz, H-7), 7.09 (1H, d, *J* = 0.5 Hz, H-2) and two protons of hydroxyl groups at δ_H_ 12.1 (1H, s, OH-8), 11.99 (1H, s, OH-1). The ^13^C NMR of **1** displayed 15 carbon signals including a methyl, two ketone groups, five methines, seven quaternary carbons (two of which are bonded to O atoms) (Table 2). Assignments of all ^1^H and ^13^C NMR signals were accomplished by interpretation of HSQC and HMBC spectra. The melting point of compound was 196 °C. The above spectral data were diagnostic of an anthraquinone having two hydroxyl groups at C-1, C-8 position, and a methyl group. Moreover, methyl group and two hydroxyl group positions were determined by the HMBC spectral correlation between the proton of methyl group H-11 (δ_H_ 2.46) with C-2 (δ_C_ 124.4)/C-4 (δ_C_ 121.4); OH-1 (δ_H_ 11.99) with C-2 (δ_C_ 124.4)/C-9a (δ_C_ 113.8) and OH-8 (δ_H_ 12.1) with C-7 (δ_C_ 124.6)/C-8a (δ_C_ 115.9). By means of its spectrometric analysis and comparison with previously published data [27], compound **1** was determined as chrysophanol or 1,8-dihydroxy-3-methylanthraquinone.

Compound **2** was obtained as orange-yellow needles. The ^1^H and ^13^C NMR spectra of **2** closely resembled those of **1**, except for the appearance of one methoxy group (δ_H_ 3.93, (3H, s), δ_C_ 56.0) and the chemical shift of C-6 (Table 2). This suggested that **2** also has an anthraquinone of **1** and **2** was the result of replacing the aromatic ring proton at C-6 in **1** by a methoxy group. This was confirmed by comparison of the NMR data of **2** with those of a known anthraquinone in Reference [28]. Thus **2** was determined to be 1,8-dihydroxy-3-methoxy-6-methylanthraquinone or physcion.

### 2.3. Quantitative Analysis of Fraxetin from Rumex Crispus Root

The content of two pure compounds chrysophanol and physcion in RCR is presented in Table 3. The quantity of these compounds was determined by the instrumentation used in this study.

### 2.4. Antioxidant Activities of the Isolated Fractions

Antioxidant activities of the isolated fractions were determined using three assays namely DPPH, ABTS free radical scavenging, and reducing power. BHT was used as a standard for all methods. The results are summarized in Table 4.

Table 4 summarizes the antioxidant activities of the EtOAc *R. crispus* root extract. For the DPPH scavenging assay, the antioxidant activity of **C2** was the highest as its IC_50_ value was the lowest (10.0 µg/mL), followed by **C3** (IC_50_ = 12.0 µg/mL). Statistically, the DPPH radical scavenging properties of **C2** and **C3** were stronger than BHT (IC_50_ = 19.2 µg/mL). The fraction **C1** (IC_50_ = 35.0 µg/mL) exhibited intermediate antioxidant capacities. For ABTS (2,2′-azinobis-(3-ethylbenzothiazoline-6-sulfonic acid)), **C2** exhibited the maximum scavenging activity with IC_50_ value of 34.3 µg/mL, followed by **C3** and BHT (IC_50_ = 44.8 and 46.9 µg/mL). The lowest antioxidant property was fraction **C1** (IC_50_ = 194.7 µg/mL). For the reducing power ability, **C2** displayed the maximum reducing power with IC_50_ value of 312.6 µg/mL. Generally, the antioxidant properties of the pure compounds **C2** and **C3** were higher than standard BHT.

### 2.5. In Vitro Inhibition of Xanthine Oxidase (XOD), α-Amylase (AAI) and α-Glucosidase (AGI)

The inhibitory effect of the isolated fractions on xanthine oxidase is presented in Table 5. The results showed that **C1**–**C3** exhibited a considerable activity against xanthine oxidase (IC_50_ = 36.4 – 88.8 µg/mL). Furthermore, **C2** manifested the highest inhibitory activity (IC_50_ = 36.4 µg/mL) followed by **C3** (IC_50_ = 36.4 µg/mL) while **C1** exhibited the least (IC_50_ = 88.8 µg/mL).

A starch-iodine method was applied to examine the inhibitory effect of isolated fractions on porcine pancreatic α-amylase. Among isolated compounds, **C3** displayed the maximum α-amylase inhibition with IC_50_ value of 113.3 µg/mL, followed by **C2** (IC_50_ = 117.3 µg/mL). The α-amylase inhibition is as follows: acarbose > **C3** > **C2** > **C1** corresponding to IC_50_ values 90.9, 113.3, 117.3, and 199.1 µg/mL, respectively.

The α-glucosidase inhibitory activity of isolated fractions was assayed using a synthetic substrate *p*-nitrophenyl-α-D-glucopyranoside (pNPG). All isolated fractions expressed higher α-glucosidase activity than that of standard acarbose. (Table 5). Of which, the activity of all isolated fractions was significantly higher than that of standard acarbose. **C3** showed the highest inhibitory activity (IC_50_ = 18.9 µg/mL) followed by **C2** (IC_50_ = 20.1 µg/mL). Fraction **C1** (IC_50_ = 91.6 µg/mL) exhibited intermediate α-glucosidase inhibition.

## 3. Discussion

In this study, two compounds were isolated and identified from the EtOAc extract of *R. crispus* root, namely chrysophanol (**1**) (fraction **C2**) and physcion (**2**) (fraction **C3**). They are biologically active compounds belonging to the flavonoid group. The radical scavenging activity of the isolated fractions was examined using DPPH, which is a frequently used method in natural product antioxidant evaluation [27], and ABTS assays. The results of this present study serve as an additional scientific finding with regards to the radical scavenging activity of the EtOAc extract of *R. crispus* root. The IC_50_ values of (**1**) (10.0 ± 0.3 µg/mL) and (**2**) (12.0 ± 0.2 µg/mL) are comparable to the value reported by a previous study on the acetone extract of the *R. crispus* root with IC_50_ of 14.0 µg/mL [28]. Moreover, a study reported that methanol extract of the fruits of *R. crispus* exhibited antioxidant activity with IC_50_ value of 3.7 µg/mL [29]. These findings strengthen the antioxidant potential of the roots of *R. crispus* over its fruits. Furthermore, this results shows that the antioxidant activities of the pure compounds (**1**) and (**2**) are higher than the standard BHT (Table 4). The results also show a strong correlation between the concentration of the pure compound (Table 3) and its antioxidant activity. Compound (**1**) (32.50 ± 0.11 µg/g DW) exhibits a stronger radical scavenging activity as seen in both DPPH and ABTS assays (IC_50_ = 10.0 ± 0.3 and 34.3 ± 0.7 µg/mL, respectively) than (**2**) (25.04 ± 0.08 µg/g DW, 12.0 ± 0.2 and 44.8 ± 0.8 µg/mL, respectively). This result is in agreement with the findings of Elzaawely et al. [30] who reported the correlation between the phenolic content and DPPH assay of the EtOAc extract of the aerial parts of *R*. *japonicus.* The same correlation is observed between the concentration of the pure compound and its reducing power, in that the higher the concentration, the stronger its reducing power.

Xanthine oxidase (XO) plays a role in gout formation since it catalyzes the oxidation of xanthine to uric acid. Compounds that inhibit the activity of XO, therefore, can be used to treat gout [31,32]. Dietary flavonoids have been reported to possess inhibitory activity against free radicals and xanthine oxidase. The antioxidant property is mainly characterized by flavonoid contents that can effortlessly release hydrogen donors to naturalize free radicals. Therefore, flavonoids could be a promising remedy for human gout and ischemia by decreasing both uric acid and superoxide concentrations in human tissues [13]. According to Mohamed Isa et al. [15], *Plumeria rubra* contains a high amount of flavonoids and could be used as a new alternative to allopurinol with increased therapeutic activity and fewer side effects. The XO inhibitory activity *in vitro* assay of methanol extract of *Plumeria rubra* flowers possesses the highest inhibition effects at IC_50_ = 23.91 μg/mL. Baicalein, baicalin and wogonin, isolated from *Scutellaria rivularis*, have been reported to exhibit a strong xanthine oxidase inhibition as evaluated by modified xanthine oxidase inhibition methods. The results showed that the order was baicalein > wogonin > baicalin, IC_50_ = 3.12, 157.38 and 215.19 μM, respectively [33]. From this present study, (**1**) (IC_50_ = 143.3 µM) exhibited a higher inhibitory effect on XO than wogonin and baicalin, whereas (**2**) (IC_50_ = 158.5 µM) inhibited XO stronger than baicalin and showed comparable inhibitory activity as wogonin. The XO inhibition activity of (**1**) and (**2**) is directly related to their concentration as extracted from the root of *R. crispus* (Table 3). Allopurinol showed a much lower IC_50_ value of 20.5 ± 0.5 µg/mL, which is why allopurinol is the preferred treatment for gout.

Both α-amylase (AA) (responsible for starch digestion) and α-glucosidase (AG) (produces glucose in the final step of the digestive process of carbohydrates) are related to type 2 diabetes since these enzymes are responsible for postprandial blood glucose levels [34]. AA and AG inhibitors are therefore widely used in the treatment of patients with type 2 diabetes, which is related to elevated postprandial blood glucose levels. Results of this present study showed consistent inhibition activities of (**1**) and (**2**) in both AA and AG in that (**2**) exhibited a stronger inhibition than (**1**) (Table 5). However, in terms of AGI, (**1**) and (**2**) turned out to be much better inhibitors compared to acarbose (Table 5). Both terpenoids and flavonoids might play crucial roles in α-amylase and α-glucosidase inhibition [35,36]. The pure compounds (**1**) and (**2**) registered a stronger anti-diabetic activity than the mixture **C3**. Moreover, the synergic effect of the two pure compounds, **C3** exerted stronger inhibition against α-glucosidase than acarbose (IC_50_ = 143.8 µg/mL). Although this study to isolate and purify active components from the root of *R. crispus*, as well as test their biological activities, was successful, *in vivo* tests should be considered in order to ascertain the anti-gout and anti-diabetic properties of this prospective plant.

## 4. Materials and Methods

### 4.1. Materials

*Rumex crispus* root (RCR) was collected (34°23′28.3′′ N 132°43′06.0” E) in October 2017. The specimen with voucher number RCR-M2017 was deposited at the Plant Physiology and Biochemistry Laboratory (IDEC, Hiroshima University, Japan). The samples were cleaned by 1% NaOCl. After blotting with tissues, samples were dried at 45 °C in the oven for one week then ground to a fine powder [37].

### 4.2. Extraction of Rumex Crispus Root

The RCR powder (1.2 kg) was soaked in MeOH (2 L) to produce 23.8 g of MeOH crude extract. The crude extract was suspended in water (300 mL), then fractionated in hexane and EtOAc to obtain 6.6, 5.8, and 9.7 g water, hexane and EtOAc extracts, respectively.

### 4.3. Isolation of Pure Compounds

The hexane extract was dried at room temperature, then was purified by MeOH to obtain fraction **C1** (200 mg).

The EtOAc extract was subjected to normal phase column chromatography (CC) using silica gel (200 g) of 70–230 mesh ASTM and LiChroprep RP-18 (40–63 mm). All fractions were examined by thin-layer chromatography (TLC) (Merck, Darmstadt, Germany). Fraction 13–15 (**C2**) (39 mg), and fraction 16–18 (**C3**) (30 mg) were crystallized after separation by CC (Figure 1). Compound **C2** (**1**) and **C3** (**2**) were obtained as pure compounds, the purity levels were confirmed by GC-MS of 98.10% and 97.79%, respectively.

### 4.4. Antioxidants Activity

Antioxidant activity of the isolated compounds was determined using DPPH (2,2-diphenyl-1-picrylhydrazyl) radical scavenging assay, ABTS (2,2′-azinobis-(3-ethylbenzothiazoline-6-sulfonic acid)) radical cation decolorization assay, ferric reducing antioxidant power test (FRAP) following previous study [37,38,39] with slight modifications.

### 4.5. Identification and Quantification of Isolated Compounds

Identification and quantification of isolated compounds was by GC-MS following previous study [40,41]. Therefore, the GC-MS system was equipped with a DB-5MS column (30 m × 0.25 mm internal diameter × 0.25 µm in thickness (Agilent Technologies, J & W Scientific Products, Folsom, CA, USA). The carrier gas was Helium and the split ratio was 5:1. GC oven temperatures operated with the initial temperature of 50 °C without hold time, followed by an increase of 10 °C/min up to a final temperature of 300 °C and holding time of 20 min. The injector and detector temperature were programmed at 300 °C and 320 °C, respectively. The MS ranged from 29 to 800 amu. JEOL’s GC-MS Mass Center System Version 2.65a was used to control the GC-MS system and process the data peak.

ESI-MS analysis was conducted on negative/positive ion mode [42]. APCI-MS analysis was implemented using a mass spectrometer with an electrospray ion source [43].

X-ray diffraction measurement was conducted with a Bruker APEX-II ULTRA diffractometer (Bruker Corporation, Billerica, MA, USA). The crystals were irradiated using Mo-Kα radiation at −100 °C. Cell parameters and intensities for the reflection were estimated using the APEX2 crystallographic software package, in which data reduction and absorption corrections were carried out with SAINT and SADABS, respectively. The structures were solved and refined with the aid of the program SHELXL. The refined structures of crystals were reported as a crystallographic information file (CIF) [44].

### 4.6. Nuclear Magnetic Resonance (NMR) Data of Chrysophanol and Physcion

*Chrysophanol*: ^1^H-NMR (500 MHz, CDCl_3_), δ (ppm), *J* (Hz): 12.1 (1H, s, OH-8), 11.99 (1H, s, OH-1), 7.82 (1H, dd, *J* = 7.0, 1.0 Hz, H-5), 7.67 (1H, d, *J* = 8.0 Hz, H-6), 7.64 (1H, d, *J* = 1.5 Hz, H-4), 7.30 (1H, dd, *J* = 8.5, 1.0 Hz, H-7), 7.09 (1H, d, *J* = 0.5 Hz, H-2), 2.46 (3H, s, H-11). ^13^C-NMR (125 MHz, CDCl3), δ (ppm): 192.6 (C-9), 182.0 (C-10), 162.7 (C-1), 162.4 (C-8), 149.3 (C-3), 136.9 (C-6), 133.7 (C-10a), 133.3 (C-4a), 124.6 (C-7), 124.4 (C-2), 121.4 (C-4), 119.9 (C-5), 115.9 (C-8a), 113.8 (C-9a), 22.2 (CH_3_).

*Physcion*: ^1^H-NMR (500 MHz, CDCl_3_), δ (ppm), *J* (Hz): 12.28 (1H, s, OH-8), 12.08 (1H, s, OH-1), 7.60 (1H, d, J = 1.0 Hz, H-4), 7.34 (1H, d, *J* = 2.5 Hz, H-5), 7.06 (1H, d, *J* = 0.5 Hz, H-2), 6.67 (1H, d, *J* = 2.5 Hz, H-7), 3.93 (3H, s, OCH3), 2.44 (3H, s, H-11). ^13^C-NMR (125 MHz, CDCl3), δ (ppm): 190.7 (C-9), 181.9 (C-10), 166.5 (C-6), 165.2 (C-8), 162.5 (C-1), 148.4 (C-3), 135.2 (C-10a), 133.2 (C-4a), 124.5 (C-2), 121.3 (C-4), 113.7 (C-9a), 110.3 (C-8a), 108.2 (C-5), 106.8 (C-7), 56.0 (OCH_3_), 22.1 (C-11).

### 4.7. Xanthine Oxidase Inhibition (XOD) Activity

The inhibitory effect on xanthine oxidase (XO) of isolated compounds (**C1**–**C3**) was measured spectrophotometrically according to the method reported previously [10] with minor modifications.

### 4.8. α-Amylase Inhibition (AAI) Assay

The inhibitory effect of **C1**–**C3** on α-amylase was assessed by a starch-iodine method [34] with modification as follows: in each well of a microplate (U-shape, Greiner Bio-one, NC, USA), 20 µL of each isolated compound was pre-incubated with 20 µL of 1 U/mL α-amylase solution (from *Aspergillus oryzae*, Sigma-Aldrich, St. Louis, MO, USA) at 37 °C for 10 min. The reaction was initiated by pipetting 30 µL of soluble starch (0.5% in deionized water). After 6 min of incubation at 37 °C, an aliquot of 20 µL of hydrochloric acid (1 M) was added to stop reaction, followed by 120 µL of 0.25 mM iodine solution. The absorbance at 565 nm was read by a microplate reader (Multiskan^TM^ Microplate Spectrophotometer, Thermo Fisher Scientific, Osaka, Japan). The inhibitory activity of **C1**–**C3** on α-amylase was performed as the inhibition percentage calculated using the formula:% inhibition = (A − C)/(B − C) × 100where A is the absorbance of the compound, B is the absorbance of reaction without enzyme, C is the absorbance of the negative control. A commercial diabetes inhibitor acarbose was used as a positive reference. Dilutions of test samples and dissolutions of enzyme used 20 mM sodium phosphate buffer (pH 6.9 comprising of 6 mM sodium chloride). α-Amylase solution and soluble starch solution were prepared and used on the day of the experiment. The IC_50_ value was calculated to exhibit 50% inhibitory activity of **C1**–**C3** against α-amylase.

### 4.9. α-Glucosidase Inhibition (AGI) Assay

The anti-α-glucosidase activity of **C1**–**C3** was evaluated using a method described previously [35] with some modifications. In brief, an amount of 20 µL methanolic stock solution of isolated compounds was pre-mixed with an equal volume of 0.1 M potassium phosphate buffer (pH 7) and 40 µL of α-glucosidase (from *Saccharomyces cerevisiae*, Sigma-Aldrich, St. Louis, MO, USA) enzyme solution (0.5 U/mL in 0.1 M potassium phosphate buffer, pH 7). After 6 min incubation at 25 °C, a 20 µL aliquot of 5 mM *p*-nitrophenyl-α-D-glucopyranoside (pNPG) substrate (in 0.1 M potassium phosphate buffer, pH 7) was added to each reaction and the mixture was incubated for another 8 min. The reaction was terminated by adding 100 µL of 0.2 M Na_2_CO_3_, and absorbance was recorded at 405 nm. The inhibition percentage was calculated by the following equation:% inhibition = (1 − A_sample_/A_control_) × 100where A_sample_ is absorbance of isolated compound, A_control_ is the abosrbance of positive controls (acarbose or quercetin).

### 4.10. Statistical Analysis

All obtained data were analyzed by Minitab Software (version 16.0, copyright 2015, Minitab Inc., State College, PA, USA). One-way analysis of variance (ANOVA) and Tukey’s post hoc test were used to identify the significant difference among mean values with *p* < 0.05. All trials were designed randomly in triplicate.

## 5. Conclusions

This study documented that the root of *Rumex crispus* possessed potent antioxidant, xanthine oxidase, α-amylase and α-glucosidase inhibitory activity in *in vitro* assays. More specifically, the compounds isolated from the EtOAc extracts emerged as a promising source of natural antioxidants, xanthine oxidase, α-amylase, and α-glucosidase inhibitors. *In vivo* tests should be conducted to affirm the bioactivity of the isolated compounds from the root of *Rumex crispus* for the development of food additives and supplements to reduce the risks type 2 diabetes and gout. The isolation of novel constituents, as well as investigations on the potent pharmaceutical properties of the root of *Rumex crispus* need to be considered.

## Figures and Tables

**Figure 1 molecules-24-03899-f001:**
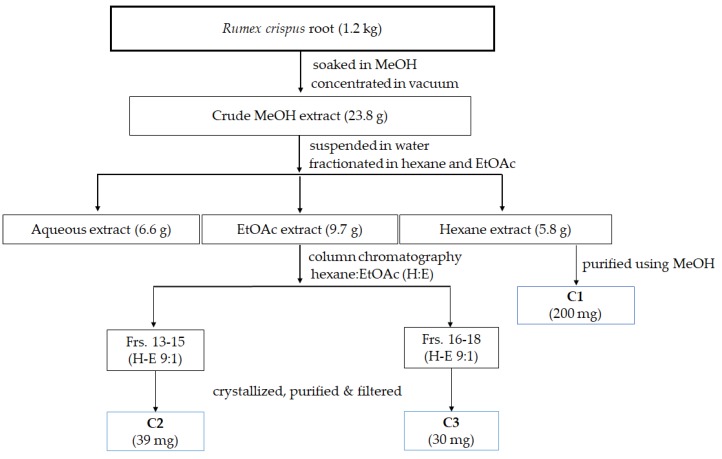
Isolation and purification of bioactive compounds from *Rumex crispus* root.

**Figure 2 molecules-24-03899-f002:**
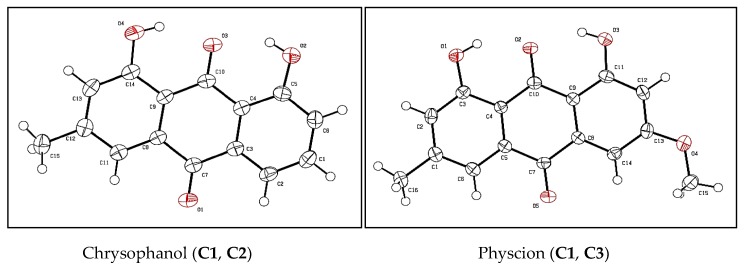
Chemical structures of bioactive constituents identified in EtOAc extract of *Rumex crispus* L. root by single-crystal X-ray.

**Figure 3 molecules-24-03899-f003:**
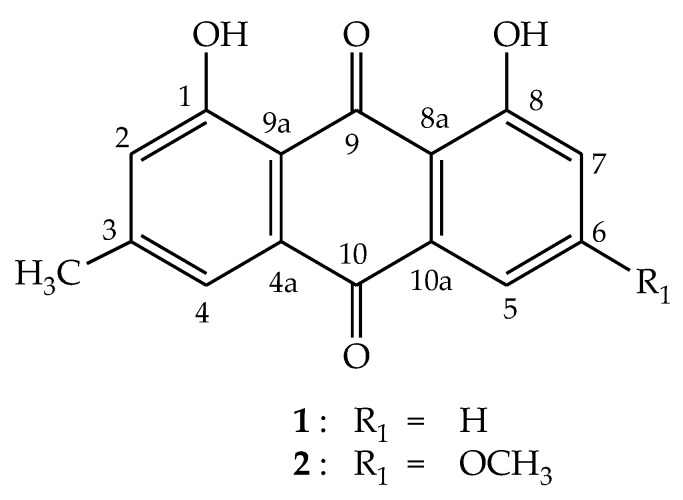
Structures of compounds **1** and **2** from *Rumex crispus* L. root.

**Table 1 molecules-24-03899-t001:** Bioactive compounds identified in EtOAc extract of *Rumex crispus* root by GC-MS.

Fractions	Retention Time	Peak Area (%)	Compounds	Chemical Formula	Molecular Weight	Similarity
**C1**	20.70	73.39	Chrysophanol	C_15_H_10_O_4_	254	90.8
	22.98	24.82	Physcion	C_16_H_12_O_5_	284	91.8
**C2**	20.71	98.10	Chrysophanol	C_15_H_10_O_4_	254	98.1
**C3**	22.99	97.79	Physcion	C_16_H_12_O_5_	284	97.8

**Table 2 molecules-24-03899-t002:** ^13^C-NMR data for compounds 1 and **2.**

Position	1		2	
	δ_C_^*^	δ_C_	δ_C_^**^	δ_C_
1	161.1	162.7	162.5	162.5
2	124.2	124.4	124.5	124.5
3	149.0	149.3	148.4	148.4
4	120.4	121.4	121.3	121.3
5	119.2	119.9	108.2	108.2
6	137.2	136.9	166.5	166.5
7	123.9	124.6	106.8	106.8
8	161.4	162.4	165.2	165.2
9	191.4	192.6	190.8	190.7
10	181.3	182.0	181.5	181.9
4a	132.8	133.3	133.2	133.2
8a	115.7	115.9	110.3	110.3
9a	113.6	113.8	113.7	113.7
10a	133.2	133.7	135.3	135.2
CH_3_	21.6	22.2	22.1	22.1
OCH_3_			56.0	56.0

δ_C_^*^ Chrysophanol in DMSO; δ_C_^**^ Physcion in CDCl_3._

**Table 3 molecules-24-03899-t003:** Quantity of pure compounds from *R. crispus* root.

Fractions	Retention Time	Compounds	Concentration
			(µg/g DW)
**C2**	20.70 ± 0.02	Chrysophanol	32.50 ± 0.11
**C3**	22.99 ± 0.05	Physcion	25.04 ± 0.08

Data express means ± SD (standard deviation).

**Table 4 molecules-24-03899-t004:** Antioxidant activities measured by DPPH, ABTS (2,2′-azinobis-(3-ethylbenzothiazoline-6-sulfonic acid)) and ferric reducing antioxidant power test (FRAP) of EtOAc extract fractions from *R. crispus* root in term of IC_50_ values.

Fractions	IC_50_ (µg/mL)		
	DPPH	ABTS	FRAP
**C1**	35.0 ± 1.6^a^	194.7 ± 2.0^a^	1366.5 ± 8.4^a^
**C2**	10.0 ± 0.3^c^	34.3 ± 0.7^c^	312.6 ± 6.3^c^
	(39.4 µM)	(135.0 µM)	(1230.7 µM)
**C3**	12.0 ± 0.2^c^	44.8 ± 0.8^b^	408.6 ± 6.8^b^
	(42.3 µM)	(157.7 µM)	(1438.7 µM)
BHT*	19.2 ± 0.3^b^	46.9 ± 0.9^b^	422.1 ± 1.1^b^
	(87.1 µM)	(212.8 µM)	(1915.8 µM)

* Positive control. Values are means ± SD (standard deviation); ^a,b,c^ indicate significant differences at *p* < 0.05.

**Table 5 molecules-24-03899-t005:** Xanthine oxidase, α-amylase and α-glucosidase inhibitory activities of isolated fractions from *Rumex crispus root* in term of IC_50_ values.

Fractions	IC_50_ (µg/mL)		
	XOD	AAI	AGI
**C1**	88.8 ± 0.9^a^	199.1 ± 1.4^a^	91.6 ± 1.4^b^
**C2**	36.4 ± 0.6^c^	117.3 ± 1.0^b^	20.1 ± 0.6^c^
	(143.3 µM)	(461.8 µM)	(79.1 µM)
**C3**	45.0 ± 0.7^b^	113.3 ± 1.3^c^	18.9 ± 0.4^c^
	(158.5 µM)	(398.9 µM)	(66.5 µM)
Allopurinol*	20.5 ± 0.5^d^	-	-
	(150.6 µM)		
Acarbose*	-	90.9 ± 0.8^d^	143.8 ± 2.6^a^
		(140.8 µM)	(222.7 µM)

* Positive control. Values are means ± SD (standard deviation); ^a,b,c,d^ indicate significant differences at *p* < 0.05.

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
