# Peer review of "Antioxidant, Xanthine Oxidase, α-Amylase and α-Glucosidase Inhibitory Activities of Bioactive Compounds from Rumex crispus L. Root"

_molecules, 2019, doi:10.3390/molecules24213899_

Round 1

Reviewer 1 Report

The paper presented for review is interesting and generally well written, however, the discussion of the results is very poor and requires deepening, especially in the part on enzymological studies. The generalities were included  without comparing the results with those obtained by other researchers.

Reviewer 2 Report

The article describes the investigation of Rumex crispus with regard to the bioactivity of some of its compounds. Phytochemicals are isolated from the plant material, characterized by standard procedures and subjected to in vitro assays, evaluating the inhibitory capacity against some enzymes (xanthine oxidase, α-amylase, α-glucosidase) and DPPH.

The experiments are correctly conducted and the results are presented accordingly. Conclusions are pertinent.

In Chapter 4.1. Materials: The number of the voucher specimen, and the Herbarium where it was deposited have to be mentioned.

However, the article has serious flaws concerning the English grammar and style, which have a strong negative impact on the readability. Many words and expressions are incomprehensible, some of them are detailed below, but the list is not exhaustive. Correct use of scientific words and expressions should be insured throughout the article prior to publication.

Line 18 : Replace “herbal plant” with “plant”

Line 20: from the ethyl acetate... (insert the)

Line 42: replace “the youthful leaves” with “the young leaves”

Line 43: replace “provocative” with the relevant disease

Line 45 : explain/correct: “driven to distinguishing proof of..”

Line 50 : explain/correct: “handle at generally low concentrations”

Line 51-52 : explain/correct: “When the oxidative stress is uncontrolled it is related with a few pathophysiological forms”.

Line 53: explain/correct: “extracts of the parting and source of R. crispus”

Line 58-59: explain/correct: ”were investigated in biological system of rules”

Line 58-59: explain/correct: “and the in-vivo consequences for a few hepatic cell reinforcement arrangement of guidelines were contemplated”

Line 86-87: explain/correct: pancreatic beta-prison cell 86 dysfunction

Line 200: replace “was extraordinarily higher” with “was significantly higher”

Line 205: explain/correct: ”The chemical group scavenging natural process of the isolated fractions”

Line 327: replace “in-vitro trials” with in vitro assays”

Reviewer 3 Report

This well written and informative paper reports results that will be of interest to members of the medicinal chemistry community.

Some minor revisions are required before publication, as follows:

The compound characterization data should be carefully checked. For example, the H-6 resonance of compound 1 should be a triplet, not a doublet as described. Melting points should be given for solids.

IC50 values should be stated in uM rather than ug/mL and should not be given for mixtures, such as extract C1. 

GC-MS retention times are given, but they have no meaning unless GC conditions and column specifications are given.

Round 2

Reviewer 1 Report

The paper was significantly improved and in its present form is suitable for publication.

Reviewer 2 Report

The authors have modified their manuscript according to the suggestions of the reviewers; the article may be published in the current form.